# Carotenoid Production from Microalgae: The Portuguese Scenario

**DOI:** 10.3390/molecules27082540

**Published:** 2022-04-14

**Authors:** Mariam Kholany, João A. P. Coutinho, Sónia P. M. Ventura

**Affiliations:** Chemistry Department, CICECO-Aveiro Institute of Materials, Campus Universitário de Santiago, University of Aveiro, 3810-193 Aveiro, Portugal; mariamkholany@ua.pt (M.K.); jcoutinho@ua.pt (J.A.P.C.)

**Keywords:** blue biorefinery, microalgae, carotenoids, downstream processes, light-harvesting pigments, Portuguese research

## Abstract

Microalgae have an outstanding capacity to efficiently produce value-added compounds. They have been inspiring researchers worldwide to develop a blue biorefinery, supporting the development of the bioeconomy, tackling the environmental crisis, and mitigating the depletion of natural resources. In this review, the characteristics of the carotenoids produced by microalgae are presented and the downstream processes developed to recover and purify them are analyzed, considering their main applications. The ongoing activities and initiatives taking place in Portugal regarding not only research, but also industrialization under the blue biorefinery concept are also discussed. The situation reported here shows that new techniques must be developed to make microalgae production more competitive. Downstream pigment purification technologies must be developed as they may have a considerable impact on the economic viability of the process. Government incentives are needed to encourage a constructive interaction between academics and businesses in order to develop a biorefinery that focuses on high-grade chemicals.

## 1. Introduction

Climate change is one of the most pressing issues our planet is facing. We must handle the requirements of a growing population while attempting to keep our civilization within the safe planetary bounds. The current reliance on a petroleum-based economy to meet the humanity’s several needs (for example, health, food, nutrition, clean energy, responsible industry) is no longer viable. In this regard, a set of calls for action—the Sustainable Development Goals (SDGs)—is based on the 2030 Agenda for Sustainable Development [1] signed by all the United Nations member states aiming to address the future humanity priorities. The bioeconomy consists of the use of materials, chemicals, and energy obtained from renewable biological resources, which closely resonates with SDGs. One of the strategies on the table is the gradual transition to the bioeconomy, which can help target and accomplish the SDGs. Seas and oceans, forests, and agriculture are three important areas of the topic of bioeconomy (related to SDG Nos. 12, 14, and 15). Algae, namely macroalgae and microalgae, but also cyanobacteria, are part of the less explored raw materials in this context. Nevertheless, microalgae are gaining importance in the framework of the bioeconomy as a carbon-neutral resource. Both its raw use and development of microalgae-based biotechnological platforms directly contribute to several other goals such as reducing poverty and hunger (SDG No. 1), promoting environmental sustainability, and developing global partnerships (SDG No. 17). 

## 2. Concept of (Blue) Biorefinery

Biorefining is a process by which the biomass is fractionated and converted into value-added products and energy in a sustainable way. Biorefinery appears as one of the most promising concepts to tackle the developing environmental and economic crisis while fighting the depletion of natural resources [2]. Considering the abundance and diversity of the biomass available, there is an opportunity to develop processes that could generate new products (Figure 1) of economic relevance. 

A biorefinery should foresee the maximization of the use of a certain biomass and the minimization of wastes. The advantage of using microalgae as the foundation of a biorefinery is that it does not clash with land and food crops. Unlike terrestrial plants, microalgae lack highly resistant cell wall components with no stem or roots, which allows for easier degradation and, consequently, easy exploitation of its valuable biocompounds [3]. Microalgae are recognized worldwide as a plentiful/valuable source of carbon of paramount industrial relevance which can be used for food, feed, chemicals, and biopharmaceuticals [4]. Microalgae have several applications in renewable sources, CO_2_ mitigation (by capturing it from the air or from industrial flows [5,6,7]) and have an outstanding capacity to produce an extensive range of bioproducts, namely polysaccharides, lipids, proteins, vitamins, antioxidants, and pigments. In addition to the rich biochemical composition, microalgae production stands out from an industrial point of view. It requires simple conditions of temperature, nutrients, and light to produce large amounts of different bioactive compounds. Under the circular economy scope, different studies have explored microalgae growth media on the basis of sewage or other liquid wastes, showing them to be efficient and profitable, while valorizing a relatively cheap waste stream [8,9,10]. Moreover, microalgae production is independent of the season, leading to overall high yields of production and biomolecules accumulation. Distinct evolutionary tactics and variable growth conditions, such as light, temperature, pH, oxygen concentration, and nutritional level, can be used to induce microalgae to produce several compounds with diverse structures and unique biological activities as surviving strategies. Thus, cultivation strategies have been adopted to maximize the production of primary and secondary unique metabolites in these cells, further boosting its industrial potential [11]. 

Thus far, the scientific community has focused on energy production when a microalgae biorefinery is addressed [3,12,13]. Microalgae are recognized as being important for the production of biodiesel, bioethanol, biohydrogen, methane, and bioelectricity, a feature extensively reviewed [14,15,16]. These fuels provide a significant potential to meet the global energy demands while providing carbon-neutral solutions and enabling CO_2_ sequestration from the atmosphere or even from industrial emissions [5,6,7]. Regardless of how efficient the current processing methods are, algae biorefineries generating biofuels alone are found to be economically infeasible. At the same time, the ability to create coproducts is a clear opportunity to make these production systems feasible [17]. In this context, hydrothermal liquefaction (HTL) has recently gained traction as a conversion technique as it allows removing the energy-demanding biomass drying step. HTL consists in the conversion of fresh biomass into biofuels by using water as the reaction medium at high temperature and pressure [18,19]. It is carried out using whole microalgae or after extraction of valuable components, converting the remainder fraction into biofuel. This second approach has the potential to broaden the array of products available from the biorefinery deserving to be more deeply studied [20].

Microalgae species, e.g., *Dunaliella salina* or *Haematococcus pluvialis*, present a great potential for specialized production of other value-added components. The complete fractionation and valorization of all the microalgae components to take the maximum advantage of the biomass is at reach and should be the goal of a microalgae biorefinery. Aiming at the maximum economic valorization of microalgae biomass, it is important to understand the market size and its value for each class of products. While microalgae-derived biofuels are attracting ever-increasing interest, many studies are showing it to be not viable economically if not followed by the valorization of the microalgae value-added compounds with far more interesting applications, namely in the pharmaceutic and cosmetic sectors, but also as fine chemicals [21]. This does not mean that sustainable resources of energy should be overlooked, but that the valorization of all biomass fractions should be considered in an integrated process. Several studies have been performed acknowledging the technoeconomic aspects of pigment recovery and a comparison of a blue biorefinery versus a traditional refinery was carried out by different authors [22,23,24]. Currently, less than 1% of the natural astaxanthin on the market comes from natural sources despite its benefits, and around 99% of astaxanthin is synthesized by big companies like BASF and Hoffman-La Roche [23]. Panis and Carreon [22] proposed a model to perform a technical and economic analysis of the production of natural astaxanthin from *Haematococcus pluvialis* in two European cities: Livadeia (Greece) and Amsterdam (Netherlands). From this study, it was concluded that natural astaxanthin still cannot compete with the synthetic one as the calculated production costs for the natural pigment (not accounting for downstream processing) were around €1536/kg (in Livadeia) and €6403/kg (in Amsterdam), while the synthetic production cost was €880/kg. Another theoretical economic study for extracting natural astaxanthin from *Haematococcus pluvialis* was published by Zgheib et al. [24]. The authors considered that the biomass suspension came from the microalgae production platform, hence, they focused on the expenses incurred in the downstream process. Economic viability was achieved with a natural astaxanthin price of more than USD 1500/kg, and biofertilizer commercialization was generated from the leftover *Haematococcus pluvialis* estimated at USD 40/kg, adding value to the process.

From the results analyzed so far, it is easy to infer that a successful biorefinery process should be designed considering all the compounds composing the biomass, from higher-value to lower. Value-added and labile molecules should be recovered first, and leftovers should be used for energy production [25]. A microalgae biorefinery may include the following steps: (i) cultivation (with optimization if required) and harvesting of cells, (ii) cell disruption to release the compounds of interest, (iii) fractionation of the compounds present in the biomass, and, in the end, (iv) purification if required to meet the demands of the area of application. The last two steps are currently recognized as the major bottlenecks for the implementation of biorefineries. Separating the various compounds without damaging the most delicate and labile bioactive fractions while keeping the high yields, low cost, and environmental efficiency is still challenging. However, there is a need for the determination of new routes to the maximum value and overall process design which can cover the use of more selective and biocompatible solvents. Among the multitude of bioactive compounds that can be recovered from this type of source, pigments should be highlighted [26]. Included in the pigment class are carotenoids (red, orange, and yellow), phycobiliproteins (red/pink and blue), and chlorophylls (green) [27], the latest being indispensable for the photosynthesis. In the next section, we discuss the principal carotenoids found in microalgae, but without being exhaustive since this information can be found in other more specialized works [23,28,29]. In this work, the ongoing activities and initiatives presently taking place in Portugal both addressing the optimization of cultivation conditions and their downstream processes not only at the research level, but also its industrialization under the blue biorefinery concept will be presented and discussed. Then, we focus more on the details of the different processes published on the extraction and purification of various carotenoids produced by microalgae. Carotenoids were the focus of this review due to their high commercial value and principal applications, a subject assessed in the end of this work. 

## 3. Microalgae Biorefineries in Portugal: Past, Present, and Future

To contextualize the scientific discussion of the previous section within the wider biorefinery approach, the Portuguese case is discussed here to highlight the ongoing efforts as well as the nonscientific hurdles to overcome. To discuss the position of Portugal on the blue bioeconomy, different sectors of activity need to be assessed. As reported in the Blue Bioeconomy Roadmap for Portugal [30], “*fisheries for human consumption and coastal tourism, shipping, shipbuilding and repair, and deep-sea shipping*” are the major activities. Moreover, “*fisheries and aquaculture, ports, processing, wholesale and retail of the main products*” are also relevant areas of development. However, and as stressed in this report [30], “*a new wave of Blue Bioeconomy subsectors has been recorded with a remarkable growth in volume of operations and projects, such as biotechnology and natural product research*”. Portugal is a maritime nation with a long Atlantic coastline, a very large exclusive economic zone (EEZ), and its history is intrinsically tied to its marine environment. The country has an immense potential to become a blue bioresource-based economy that can meet current global economic, environmental, and societal demands in the future. As discussed before, marine resources are widely acknowledged as the basic materials for a wide range of applications that extend far beyond food and feed production. Novel drugs, cosmetics, nutraceuticals, and chemicals are emerging at an outstanding rate. Moreover, marine resources are multisector, cross-cutting through the value chain. The blue bioeconomy value chain accounts for all stakeholders, from those who harvest and/or produce the biomass to product developers and distributors who deliver blue bioproducts or services to the end user. It can be easily broken down into the four main areas: biomass production and/or harvesting, innovation and development, commercialization and market entrance, and support services. Biomass production and harvesting are performed by researchers interested in biodiversity or bioprospecting aquatic organisms, fishery actors harvesting wild organisms, and aquaculture companies. This is followed by innovation and development activities, namely the transformation of the biomass into value-added products, the development of innovative applications, and the development and implementation of pilot-scale technologies and infrastructure. The focus is on the development of numerous marine-based products, such as bioactive compounds, pigments, nutrients, and enzymes. Fractionation and refining processes are also considered in the context of waste stream valorization by transformation of certain byproducts into value-added chemicals for other industries. This activity targets commercialization and market entrance, which include product manufacturing and processing, as well as final product creation and distribution. Support services (from public administration to marketing, legal, and funding entities) and research and development are transversal to all the previously mentioned steps of the value chain.

### 3.1. Blue Biorefinery Initiatives and Activities in Portugal

Traditionally, in Portugal, blue bioeconomy activities have been concentrated on human consumption fisheries and coastal tourism [31]. Nevertheless, biotechnology and natural product research are two emerging subsectors in the blue bioeconomy that have seen a significant increase in the number of operations and initiatives [32]. The Portuguese blue bioeconomy has been pointed out as one of the most promising sectors of the economy. This has been a recurring topic on political agendas, as well as a priority for public funding distribution. Several European funding mechanisms covering marine-related activities as well as national funding schemes such as MAR2020 and Fundo Azul have been created to incentivize this sector in Portugal in the past years. Several scientific projects have received financial support through European funding sources. From 2015 onwards, approximately 11 million euros have been allocated to Portuguese partners through European H2020 calls targeting the blue bioeconomy, marine biotechnology, and marine bioresources [33], although only 7% of the attributed funding have gone to projects coordinated by a Portuguese partner. The recognition of the critical role of the ocean and the blue bioeconomy in Portugal has sparked the development of numerous new studies and strategies, such as the Thematic Agenda for Research and Innovation for the Sea 2030 [33] produced by a group of Portuguese agents under the coordination of Fundação para a Ciência e Tecnologia or the PorTECH Cluster strategy launched by the Ministry of the Sea. 

Small and medium-sized enterprises (SMEs) account for a high share of Portuguese entities involved in the blue bioeconomy, underlining how the private sector plays a key role in the Portuguese blue bioeconomy. The most well-represented entities after SMEs are academic research institutions, incubators, associations, and nongovernmental organizations (NGOs). Among these, a new type of private, nonprofit organizations is projected to be critical—collaborative laboratories (CoLabs), composed by academia and companies, whose main objective is to implement research and innovation agendas and promote technology transfer from the R&D sector towards companies to generate economic and social value. These organizations encourage the development of collaborative practices between scientific, technological, and higher education institutions, as well as the social and economic sectors.

### 3.2. Main Sources of the Marine Biomass in Portugal

The main resources recognized for blue biorefineries encompass microalgae, cyanobacteria, bacteria, fungi, and other microorganisms, macroalgae, sponges, mollusks and other invertebrates, fish, and fish industry byproducts. According to the Blue Bioeconomy Roadmap for Portugal (Figure 2), 31% of the Portuguese stakeholders use algae (micro- and macroalgae, at 16% and 15%, respectively); 36% focus their activities on microorganisms (cyanobacteria, bacteria, fungi, and others); and 37% of the Portuguese actors focus their activities on fish. Both microorganisms and algae are used mostly by SMEs and academic research entities.

### 3.3. Main Areas of Application

A substantial portion of the Portuguese blue industry is concentrated on food applications. Nevertheless, the sectors of blue development extend to the food, feed, pharmaceutical, nutraceutical, cosmetic, medical, biomaterial, antifouling, bioplastic, and textile areas. Stakeholders have expressed a strong desire to develop new fields of application during the next ten years [30], resulting in a more evenly distributed field of application in the future. This is especially relevant for applications involving bioplastics and biomaterials, which currently lead the blue biotechnology research and development, owing to the current environmental concerns and societal demands for more sustainable materials. According to experts, researchers, producers, and stakeholders, the production of proteins for food and health applications, including vaccines and recombinant proteins, the production of polysaccharides and antibiotics, and the use of microalgae for bioremediation and biofertilization are the most promising markets expected for microalgae in the coming decades [34,35].

A4F (Algae for Future) (https://a4f.pt/pt, accessed on 16 February 2022) is one of the main Portuguese microalgae producers, with several operations running. A4F is responsible for the production of microalgae while also supporting the development of new products and applications in several areas. ALGATEC—Eco Business Park (https://a4f.pt/pt/projetos/algatec-pt, accessed on 16 February 2022) is a big research center and pilot facility managed by A4F in collaboration with Solvay [36]. This park brings together several microalgae agents involved in production, textile, biomaterials, medicines, nutraceuticals, and bioremediation, to name just a few. Other established agents can be highlighted, namely AllMicroalgae (https://www.allmicroalgae.com/en/, accessed on 16 February 2022), Sparos (https://www.sparos.pt, accessed on 16 February 2022), Necton (https://necton.pt, accessed on 16 February 2022), and Buggypower (http://www.buggypower.eu, accessed on 16 February 2022), working across the food, feed, pharmaceutical, nutraceutical, and cosmetic fields. 

### 3.4. Challenges Faced by the Sector in Portugal

Despite its enormous potential, Portugal continues to face various hurdles that stymie its progress and exponential growth towards the blue bioeconomy. To illustrate this, data on wild-harvested species and those that may be cultivated and mass-produced in controlled conditions within the Portuguese jurisdiction areas, which would be a key starting point for the blue biorefinery pipeline, are still lacking and are often disregarded. This information is vital to maximize the potential of any venture in this field. As such, mapping such issues is critical for proposing solutions to overcome those obstacles and promote the development of the blue bioeconomy in Portugal.

The obstacles to the advancement of the blue bioeconomy mentioned by national actors were summarized into the following categories: Science, technology, and logistics, linked to insufficient knowledge, training, scientific development, or capacity to create, improve, scale up, transport, and accommodate perishable (marine) raw materials or to the ability to implement products, services, or processes throughout the value chain. For instance, securing or developing a reliable, replicable, continuous, and sustainable biomass source based on marine resources is the most severe bottleneck jeopardizing the development of marine-derived pharmaceuticals.Cooperation, associated with limited knowledge transfer and communication between the national agents holding the data, knowledge, or infrastructure that could promote innovation or overcome barriers impeding the creation or development of a product or service is among the most severe challenges for Portugal.Communication and marketing, related with communication throughout the value chain, together with the need for marketing skills and expertise should be improved.Market and consumer demand, addressing the knowledge on the development and implementation of business plans that address market and consumer demands as well as competitor products.Funding and cost of operations, referring to available funding schemes, access to them and their appropriateness, along with the development of cost-efficient processes. Here, it is important to highlight how access to blue resources is often difficult or expensive and though specific public funding schemes may propose a swift solution, these are generally not suitable as a continuous stream of reliable funding.Legal and regulatory, connected to licensing and regulation agendas required to start new enterprises, develop and commercialize new products or services while addressing intellectual property issues. Hurdles in dealing with public authorities or applicable regulatory authorities are also included. This is frequently cited as the deciding reason for failing to attract private investment due to stringent Portuguese rules.

Concerning this last point, we could cite the obstacles faced by the introduction of new products and solutions derived from novel microalgae or macroalgae due to the EU list of authorized food components and production standards [37]. However, these safety compliance standards are often not applied to non-EU-produced items that compete unfairly with those developed in the EU. It is paramount that legal and regulatory organizations keep up with the processes of blue bio-based innovation and development.

### 3.5. Actions towards the Blue Bioeconomy

To overcome some of these challenges, actions need to be taken, which can only be accomplished by a widespread collaboration among partners. It is vital to encourage innovative and varied economic initiatives that capitalize on our excellent scientific and technological innovation capabilities, extensive national jurisdiction areas, and international transportation of products and services. A well-organized value chain can stimulate both international economic growth and the domestic market. 

A thorough plan was prepared and a series of short-, medium-, and long-term actions are detailed in the Blue Bioeconomy Roadmap for Portugal [30], where actions and timetables are proposed to promote each sector. Most measures proposed are transversal to the value chain. Briefly, the overall analysis of all the actions and implementation of the roadmap can be summarized as follows:Public support for the creation of a web-based portal to gather and group information on all the blue bioeconomy-concerned entities, their expertise, products, and services.Development of an infrastructure to centralize demand for blue bioresources, prototyping and pilot-scaling up facilities, such as downstream processing and biorefineries. The Blue Demo Network (initiative promoted by the BLUEBIO ALLIANCE) could promote this action through its continuous financial support.Revision of training programs for young scientists in blue bio-based courses to include training sessions targeting industry and economy demands, namely entrepreneurship, management, and industrial skills.Simplification of national funding schemes, through the submission process for projects exerted in two stages: the first, simpler version, followed by a full project proposal submission conditioned by approval in stage one, similar to EU instruments such as the SME Instrument [38]. This can potentially reduce the time invested in such complex funding applications, as well as the time and financial resources for the evaluation.Creation of blue bioeconomy-related acceleration and follow-up programs. An open innovation project calls in and the industry challenges stakeholders throughout the value chain to address their needs through competitive calls and funding. Initiatives for funding high-risk experimentation and exploratory projects should also be promoted.

It is worth noting that some of the suggested initiatives are being addressed by a recent program launched by BBA—the Blue Demo Network. 

Portugal possesses the prerequisites and resources to become one of the main centers for the emergence of the blue bioeconomy due to its vast pool of natural resources and well-trained and highly skilled scientists combined with its extensive industrial knowledge. 

There is a potential to create a blue-based model that adds economic value while respecting natural assets. By employing a combined strategy and attracting national and international investment, new technology-based industries and jobs will be created, which can provide the needed impetus in Portugal in a relatively short timespan.

## 4. Carotenoids Found in Microalgae

Carotenoids are ubiquitous light-harvesting pigments in nature and can be found in higher plants, fruits and vegetables, macroalgae and microalgae. The global market for carotenoids was approximately $1.5 billion in 2017 and should have reached $2.0 billion by 2022 [39]. There are over 1100 carotenoids identified in living organisms [40]. More than merely pigmentation, these molecules serve important, often critical, roles in biological systems. Most carotenoids have therapeutic value, including anti-inflammatory and anticancer activities, which are attributed to their high protective action against photo-oxidative damage to cells [28,41].

Carotenoids are lipophilic pigments with isoprenoid structures characterized by a polyene backbone of conjugated double bonds. Both the pigmenting properties and their ability to act as antioxidants (by means of an interaction with free radicals and singlet oxygen) are accounted for by this structure. Differences on this polyene backbone, such as the number or position of the conjugated double bonds together with the addition of oxygen functional groups affect the reactivity of the carotenoids. Thus, carotenoids can be categorized into two classes: (i) carotenes, hydrocarbon hydrophobic carotenoids (e.g., α-carotene, β-carotene, lycopene) (ii) and xanthophylls, polar oxygenated derivatives (e.g., lutein, zeaxanthin, astaxanthin). Xanthophylls are usually restricted to light-harvesting complexes (performing both light capture and photoprotective roles), whilst carotenes can be found in reaction centers considering their protective role [42,43,44].

Microalgae are considered one of the best commercial sources of natural carotenoids, with a focus on strains of the Chlorophyta phylum, such as *Dunaliella salina*, *Haematococcus pluvialis*, and various *Chlorella* sp. [28,45]. *Dunaliella salina* is well-known for its high production and accumulation of carotenoids. For most microalgae, the average concentration of carotenoids is ca. 0.1–2%, yet under specific stress conditions of hypersalinity and light intensity, *Dunaliella salina* can produce up to 10–14% of its dry mass [46,47]. The fact that this unicellular alga lacks a rigid cell wall while being able to be cultivated outdoors in open ponds (owing to the extreme conditions under which it grows) lends itself to industrial production. 

More than 95% of the carotene content of microalgae is accounted for by β-carotene as a mixture of *trans* and *cis* isomers. The remainder of the carotene content includes, but is not limited to, α-carotene, lutein, zeaxanthin (Figure 3), and this distribution may vary for different strains [48,49,50,51,52].

### 4.1. β-Carotene

Without any doubt, β-carotene is one of the most relevant carotenoids. This pigment has a crucial role in human health as it is an active form of provitamin A, an additive to nutritional health products; β-carotene can be applied in several industries, with applications in food, feeds, pharmaceutics, and cosmetics. The increasing demand for natural carotenoids leads to the growing interest in the extraction and purification of β-carotene from different natural sources. The natural β-carotene occurs both in the all-*trans* and 9-*cis* isomers, contrarily to the synthetic β-carotene, where only the all-*trans* isomer is present [53]. The natural 9-*cis* isomer contributes greatly to the quenching of free oxygen radicals, protecting the cell from oxidative damage. Nonetheless, an all-*trans* isomer is twice as active in the vitamin A formation. The consumption of β-carotene is related to the prevention of UV-induced erythema in humans. Moreover, β-carotene intake is linked to the inhibition of low-density lipoprotein (LDL) oxidation and plasma triglyceride levels, cholesterol, and high-density lipoprotein (HDL). Additionally, according to several epidemiological studies, a diet rich in β-carotene preserves the average serum levels and lowers the incidence of several types of cancer and degenerative diseases [11,54].

### 4.2. Lutein and Zeaxanthin

Lutein and zeaxanthin are dihydroxy dicyclic isomers derived from α- and β-carotene, respectively. The carotenoids differ solely in the placement of a single double bond, resulting in lutein having a β-ring and an ε-ring, and zeaxanthin having two β-rings. This difference results in a light-yellow color for lutein, whereas zeaxanthin has a darker yellow color. These pigments accumulate in the macula of the human retina, alongside meso-zeaxanthin (which is partially derived from lutein) and are collectively termed the macular pigment [55]. There is a growing body of proof regarding the importance of lutein and zeaxanthin in eye health, although stronger evidence is still required to support that they may reduce the risk of developing age-related macular degeneration (AMD) or slow the progression to late-stage AMD [56]. For this protective role, the two carotenoids act on two fronts: (i) absorbing damaging blue light, acting as filters; (ii) and as antioxidants quenching singlet oxygen and scavenging harmful reactive oxygen species [57]. These carotenoids are naturally produced by the microalgae showed in Table 1. Microalgae such as *Porphyridium cruentum*, *Spirulina maxima*, and *Dunaliella salina* might be of great value in the production of various types of these carotenoids. They present very interesting biological activities and have been studied and applied in various fields (Table 1). To prevent oxidative damage and disease conditions arising from such damage, a combination of carotenoids possessing different chemical characteristics can be used. 

## 5. Cell Disruption, Extraction, and Purification Techniques

Notwithstanding the high economic value of some of the biocompounds present in microalgae, their commercialization is still limited. The development of new products from microalgae bioresources may involve several steps. In spite of the perceived advantages of carotenoids, the large scale and cost-effective downstream processing is still quite challenging, particularly while maintaining high extraction yields and cost and environmental efficiency. However, if a good characterization of the biomass structure is performed, an opportunity for an efficient biorefinery arises. Downstream processing of the biomass is normally associated with the three main steps: (i) cell disruption, (ii) extraction; (iii) and separation/purification. 

### 5.1. Cell Disruption

The cell disruption step can be critical to the successful downstream processing of microalgae. Microalgae cells have a multilayered cell wall, and compounds of interest are typically bound to cell membranes or found as globules at the core. As a result, the cell wall serves as a physical barrier against solvent permeation [65]. Cell disruption techniques are employed to disrupt or disintegrate the barrier, thus increasing the recovery yield of the desired components. There are several techniques that can be applied with varying degrees of efficiency depending on the cell wall characteristics of a given microalgae species [65]. Some microalgal cells are easily broken through gentler or more energy-efficient disruption techniques. Other cells may demand a harsher and more thorough disruption method [66].

The available methods for the release of the intracellular content can be divided into two main groups: (i) mechanical, and (ii) nonmechanical techniques [67]. These methods can be combined or used alone and present their advantages and disadvantages considering the energy consumption, compatibility with other downstream operations and the process economy of alternative strategies. 

Mechanical and physical techniques induce disruption of the cell wall through mechanical forces, namely solid shear forces (e.g., bead mill, high-speed homogenization), liquid shear forces (e.g., high pressure homogenization, microfluidization), energy transfer through waves (e.g., ultrasonication, microwaves), and currents (e.g., pulsed electric field) [66]. These techniques are generally easy to scale up and allow for a higher recovery of intracellular compounds. Their broad application motivated the understanding of these mechanisms and has been well-reviewed [66]. Nevertheless, some downsides can be emphasized, namely its nonspecific character. Moreover, the harsh conditions used during mechanical cell disruption can negatively affect the biological activity of the target molecules and the downstream process due to the fine cell debris caused, and the high-energy requirement limits scale-up as careful control is needed to remove the excessive heat generated to avoid degradation. The high energy consumption is another negative point that must be highlighted. 

Nonmechanical techniques are generally considered more selective and gentler. Here, cell lysis is achieved with chemical agents (e.g., antibiotics, chaotropes, hypochlorites, chelating agents, acids and alkali, detergents and solvents), enzymatic treatments, osmotic shock, and temperature (e.g., thermolysis, autoclaving, freeze-thawing) [65,66]. These methods often only induce perforation or permeabilization of the cell wall rather than shredding, such as through specific interactions with the wall that allow the product to leach out. Most studies on the microalgae biorefinery use solvents or surfactant solutions for the extraction of target molecules (e.g., pigments, proteins, and carbohydrates amongst others). However, tensioactive solvents and surfactant solutions can cause distortions on the cell wall or membrane and consequently lead to cell rupture. Thus, solvent-induced disruption may be considered without other pretreatment of the biomass depending on the composition of the microorganism. These methods are often limited to the laboratory scale owing to the low efficiency and economic limitations since they may require additional steps in the downstream process [4]. 

Table 2 summarizes the main cell disruption methods, highlighting the principle of cell disruption and the main advantages and disadvantages of each. The efficiency and suitability of the cell disruption method are dependent on the microalgae species, the cell wall composition, and the properties of the target products. Nevertheless, combining two or more cell disruption methods, such as the combination of aqueous surfactant solutions with ultrasonication [68], is a way to increase cell disruption efficiency and, consequently, the recovery of the target compound. 

### 5.2. Extraction

As previously stated, microalgae are composed of a variety of molecules of interest with commercial value that must be separated from the remaining cellular components. Several extraction methods can be applied after a cell disruption step or applied directly to the cell. Solvent extraction methods include the use of organic solvents, surfactant solutions, ionic liquids, deep and supercritical eutectic solvents, among others. The majority of the research has targeted lipid recovery from microalgae [80,81]. Nevertheless, in recent years, numerous approaches have been explored to include different classes of molecules (namely, pigments, carbohydrates, or proteins) [13]. Carotenoid extraction has to be tailored in accordance to the hydrophobicity of the molecule, and several techniques have been proposed for this task. Table 3 comprises a list of extraction techniques utilized for pigment recovery, namely of carotenoids, while providing examples of application. 

#### 5.2.1. Organic Solvents

One of the most well-established techniques for the extraction of biomolecules, mostly lipids, from microalgae is the use of organic solvents. This type of extraction is often preceded by the cell disruption step to facilitate solvent penetration and access to the cell core, as it significantly improves the effectiveness of extraction, increasing the recovery yield of the desired components [89]. Different approaches can be taken with the traditional solvents, namely the solid–liquid extraction method, the Bligh and Dyer method, the Folch method, and the Soxhlet method [90]. The most commonly used organic solvents are hexane and methanol, but also benzene, acetone, chloroform, methylene chloride, and isopropanol or a combination of organic solvents depending on the products desired [91,92]. This approach tends to focus on lipophilic compounds, neglecting the remaining compounds of interest. To this end, nonpolar organic solvents are more selective to lipid extraction, while polar solvents tend to dissolve lipids with other microalgae components, such as carbohydrates, pigments, or amino acids. Zou et al. [82] investigated the effect of ultrasound with an ethanol and ethyl acetate solution (1:1), finding that the yield of astaxanthin was 27.58 ± 0.40 mg·g^−1^ after ultrasound irradiation of 200 W for 16 min. The impact of pretreatment induced in the solvent aided cell disruption. This allowed for more liquid phase penetration and diffusion into the rigid cell wall. The standard solvent extraction procedure with the same solvent mixture yielded a lower yield of astaxanthin (17.34 ± 0.85 mg·g^−1^) and took longer to process. Although the extraction by organic solvent provides benefits, such as low cost and ease of scale-up, the prolonged processing time and the use of large amounts of organic solvents make this extraction technique undesirable for future biorefineries. Moreover, the high vapor pressure of most organic solvents allows for their easy distillation, recovery, and separation from the cell residue, thereby facilitating downstream processing whilst minimizing solvent loss. As such, an economic and environmental analysis on the whole process should be considered before ruling out an organic solvent based on its toxicity alone.

#### 5.2.2. Surfactant Solvents

Surfactants are a wide group of amphiphilic molecules consisting of a hydrophilic “head” and a hydrophobic “tail”. Above the critical micellar concentration (CMC), aggregation structures called micelles are formed. In aqueous media, to reduce the unfavorable contact with water, hydrophobic regions are directed towards the core of the micelle, while hydrophilic parts are in the periphery to maximize the contact with the aqueous media. Surfactants can be classified as anionic, cationic, nonionic, and zwitterionic according to the charge of the polar head group [93]. Above their CMC, it is generally possible to observe that surfactant molecules can intrude the cell membrane, thereby changing the bilayer physical properties, including permeability, and leading to a cell membrane breakdown. Aqueous solutions of surfactants take advantage over other solvents more commonly used in that they require lower concentrations, eventually leading to cheaper and more sustainable processes by valorizing water as the solvent. Several studies have been performed, taking advantage of these significant characteristics, to recover different molecules [94,95,96]. Dorado et al. [84] proposed a process that simultaneously extracts, encapsulates, and stabilizes astaxanthin from *Haematococcus pluvialis*. This novel approach takes advantage of the combination of supramolecular amphiphilic solvents and nanostructured lipid carriers. The surfactant-based nanosystem consisting of octanoic acid, ethanol, and water allowed for a maximum yield of extraction of 96 ± 7%.

#### 5.2.3. Ionic Liquids and Deep Eutectic Solvents

Ionic liquids (ILs) are salts bearing low-charge density and low symmetry among their ions, typically composed of a large organic cation and an organic or inorganic anion, which leads to decreased melting points in comparison with the common salts. ILs can be composed by a vast array of ions that, carefully selected, can allow a wide range of structural combinations—helping to develop their recognition as *designer solvents* [97]. Beyond their unique properties arising from their ionic nature, ILs provide a tunable solvation environment capable of accommodating solutes of ranging polarity with variable selectivity through the modulation of specific interactions such as hydrogen bonding, π⋯π, van der Waals and Coulombic interactions [98]. Deep eutectic solvents (DES) are often considered as an economic alternative to ILs and as such have been drawing a lot of attention in the last few years. DESs are a mixture between two (or more) starting materials, a hydrogen bond acceptor (HBA) and a hydrogen bond donor (HBD), where the eutectic temperature of the mixture is considerably lower than the melting temperature of either of the constituents [99,100]. Analogous to ILs, this class of solvents can be designated as *designer solvents* since, and again, it is possible to explore numerous combinations of HBA and HBD, with tunable properties to selectively dissolve and extract biomolecules of interest from the biomass.

The application of ILs or DESs as alternative solvents to extract biomolecules from the biomass is well-established [98]. ILs were applied to the extraction of pigments [101], alkaloids [102], proteins [103], terpenoids [104], and lipids [105]. Moreover, the application of ILs to the extraction and fractionation of bioactive compounds from microalgae with high industrial/commercial interest was already reported [106,107]. Similarly, DESs were employed for the extraction of a variety of biomolecules from the biomass, namely of lignin [108], phenolic compounds [99,109], proteins [110], and flavonoids [111]. Furthermore, DESs were proposed for extraction processes from microalgae [112,113,114]. Nevertheless, significant research gaps remain in the application of IL- or DES-based extraction processes of other compounds from microalgae, the most significant being the product recovery and the solvent regeneration. Contrary to organic solvents, the low vapor pressure of ILs and DESs prevents their economical distillation, which in turn can complicate the downstream purification processes. In addition to liquid–liquid separation detailed further on, alternative purification techniques for pigment recovery include the selective crystallization of the IL [101] or through resin adsorption [115].

ILs or DESs might allow a more sustainable process due to the increased efficiency of the process, higher biocompatibility of the compounds being extracted, and the decrease in the economic impact (through IL recyclability, for instance). However, given their possible toxicity and environmental persistence, the choice of an IL or DES must be critically investigated [116,117,118,119]. To counter some of their drawbacks, the preference for their aqueous solutions should be considered since IL aqueous solutions have been shown to allow higher yields of extraction due to a decrease in their viscosity and the higher solubility of compounds in aqueous solutions (due to the hydrotropic nature of ILs in water) [120,121]. Moreover aqueous solutions of ILs favorably affect the environmental and economic impacts of the downstream process [98]. Another aspect to be considered is the use of more benign ILs, allied with their recycle and reuse. The global concern for more benign yet cheaper solvents is driving researchers to select cholinium [122], amino acids [123], and derivatives of mandelic acid, imidazolium and pyridinium, IL-based cations of lower toxicity and higher biodegradability [124]. This feature significantly broadens the ILs’ applicability and it has attracted great interest from the modern scientific community, especially for extraction and separation of biocompounds. 

#### 5.2.4. Supercritical Fluid Extraction

Supercritical fluid extraction (SFE) is becoming progressively popular as an extraction process, particularly in the food and pharmaceutical sectors. SFE combines extraction and separation by tightly managing several process parameters and conditions. Supercritical fluids (SCFs) are substances that exist at temperatures and pressures above their critical point and effectively combine the features of liquids and gases [125]. The most commonly used supercritical fluids are carbon dioxide (scCO_2_) and water, sometimes modified by cosolvents such as ethanol or methanol. Supercritical fluids possess several unique traits in this state, namely, lower viscosity than in liquids, the capacity to dissolve better than gases, and more diffusive power. Concerning the extraction of biocompounds from microalgae, this technique has been applied to the recovery of carotenoids and other pigments [126,127], as well as of polyunsaturated fatty acids (PUFAs) [128].

### 5.3. Fractionation/Purification

As previously discussed, the downstream processes applied to the biomass as a raw material normally comprise the three main steps: (i) cell disruption, (ii) extraction, (iii) and separation/purification, the latter being fully dependent on the final application demands. Even if highly efficient, extraction processes, particularly if applied after the cell rupture step, are typically associated with low selectivity, meaning that the obtained biomolecule-enriched extracts show low purity. Depending on the final application of the extracts, and for pharmaceutical/medical-oriented products in particular, highly purified carotenoids are required, although the downstream schemes may be more expensive. Moreover, there is a growing interest of the industrial and academic communities in marine resources (one of the pillars of The Agenda 2030 for Sustainable Development) that has been charming researchers to develop new and more effective carotenoid-oriented downstream processes. The need for new purification methods and the establishment of industrially efficient alternative solvents and methods is, thus, vital.

The following description of fractionation techniques focus on the ones primarily used for carotenoid recovery. As such, techniques for protein fractionation are only briefly discussed as these are usually removed in the first separation step (e.g., induced precipitation and ultrafiltration). It is important to note that no single step of fractionation is totally selective and usually requires a multi-separation process design targeting a specific compound at each stage that is dictated by the choice of the extraction methodology.

#### 5.3.1. Protein Removal

The precipitation of aqueous solutions of proteins through the addition of ammonium sulfate is a very well-established and economical method, concentrating the carotenoids in the liquid phase. By adding a salt into an aqueous solution containing the proteins, the salt dissociates and the water that once offered a great solvating power to proteins solvates the dissociated ions preferably. Thus, the charges of the protein molecules tend to interact more, forming aggregates and precipitates, an effect called salting out. In addition, ammonium sulfate is usually chosen for this purpose due to its high water solubility and low cost [129]. 

Membrane techniques allow various substances to be separated based on their size or charge. Briefly, molecules bigger than the pore size of the membrane are retained in traditional membrane filtration, while lower molecules flow through the membrane pores regardless of their nature. Ultrafiltration is a popular method for concentrating macromolecules and exchanging buffers. High retention of proteins and enzymes, for example, can be accomplished by utilizing decreased pore size membranes and moderate temperature. However, unlike other techniques, UF has no selectivity and faces problems of fouling and target molecule absorption. Nonetheless, it is a low-cost technology that is simple to scale up for commercial uses [130]. Electromembrane filtration introduces separation by charge and molecular weight without the need for pressure, making it more selective than conventional membranes [131].

#### 5.3.2. Chromatography

Chromatography is one of the most common purification methods for biomolecules. There are several chromatographic techniques that have been explored, namely ion exchange chromatography, hydroxyapatite chromatography, gel filtration, and expanded bed absorption chromatography. Chromatographic methods separate molecules in a mixture based on differences in their properties, such as adsorption (liquid–solid), partition (liquid–solid), affinity, and molecular weights [132]. There are various types of chromatography, and the effectiveness of each is determined by the nature of the molecule to be separated. Several works highlight the purification of carotenoids using chromatographic techniques [133,134,135]. Although extremely efficient in the partition of different molecules, chromatography generally suffers from high costs linked to chromatographic columns, low loading capacities, and poor scale-up opportunities. Thus, these chromatographic techniques are mostly only used for the purpose of product purification with a considerably high value in the pharmaceutical/cosmetics industries [136]. 

#### 5.3.3. Liquid–Liquid Extraction

Liquid–liquid extraction (LLE) is based on the partition of a solute between two phases composed of two immiscible or partially immiscible liquids, usually water and an organic solvent [137]. Two-phase aqueous–organic systems have been proved to be an effective way to separate poorly water-soluble compounds, such as lipids and carotenoids. In these systems, a biocompatible organic solvent is in contact with the aqueous phase where partition occurs based on hydrophobicity. The high hydrophobicity of carotenoids makes them good candidates for this technique. In spite of LLE being cost-effective, quick, simple, versatile to operate, and easy to scale up, allowing the recycling of solvents, the use of high amounts of toxic and volatile water-immiscible organic solvents of poor selectivity is the main drawback to be overcome [138]. Oil/IL/water systems were investigated for the extraction and separation of chlorophyll and fucoxanthin from the brown algae *Saccharina latissima* (Linnaeus), with yields of extraction of chlorophyll and fucoxanthin achieving 4.93 ± 0.22 mg_chl_·g_dry biomass_^−1^ and 1956 ± 84 μg_fuco_·g_dry biomass_^−1^ at the system composition of 84% of IL (350 mM) + 16% of sunflower oil [139].

#### 5.3.4. Aqueous Biphasic Systems

Aqueous biphasic systems (ABS), a particular type of LLE, appears to be an appealing alternative. ABSs retain the benefits of traditional LLE while adding the advantages of biocompatibility and tunability [140]. ABSs are based on two water-rich phases, which are formed by mixing two incompatible water-soluble solutes above certain concentrations. One of the aqueous phases is enriched in one of the solutes, while the coexisting phase is mainly formed by the other main solvent. Although pairs of two polymers and of a polymer and a salt are the most commonly used [141], multiple combinations have appeared in the last decades. Some examples include the use of carbohydrates [142], amino acids [143], ILs [144], and surfactants [145]. ABSs have already been shown to be highly efficient and selective if well-selected, from the extraction of small compounds (alkaloids, phenolic compounds, and amino acids) and drugs from pharmaceutical wastes or wastewaters to more sensitive biomolecules (proteins, antibodies, and nucleic acids) from natural matrices [141]. Due to their water-rich environment, ABSs are recognized as a more suitable technology for the recovery and purification of labile biomolecules [140], and they also present a good potential to be scaled up [146,147], allowing the development of integrated extraction and purification processes. The performance of an ABS is dependent of numerous factors, such as the affinity of the biomolecule for each phase and type of interactions between the biomolecule and the components of the system. Particularly, Cisneros et al. [148] proposed a polyethylene glycol (PEG)–phosphate ABS for the recovery of lutein from the green microalga *Chlorella protothecoides*. In that study, a pigment recovery of 81.0 ± 2.8% was attained. Furthermore, it is possible to integrate multiple steps with this technology upon choosing the phase-forming solvents. ABSs formed with surfactants, for instance, allow bridging the gap between hydrophobic and hydrophilic systems, enabling the integration of cell disruption, extraction, and separation. Kholany et al. proposed an integrated process in which violacein was extracted from the biomass by using aqueous solutions of a surfactant (Tween 20), followed by its separation from the main contaminants by applying an ABS composed of Tween 20 + [Ch] [Ac] + water [149]. 

The potential benefits of ABSs include high-performance extraction as well as quick phase separation and low viscosity favoring mass transfer [150]. In biotechnological processes, the performance of a technology should not be the unique point to consider. In fact, sustainability and environmental subject matters should also be addressed.

#### 5.3.5. Integrated Techniques—Example

The efficacy and applicability of the aforementioned fractionation techniques are dependent on different factors. To obtain a higher purity of the desired compounds, a multi-separation process design may be necessary. An example of this is the combination of LLE and the chromatography technology in centrifugal partition chromatography (CPC), or continuous tubular separators (CTS). These technologies allow scaling up the LLE- or ABS-based purification processes and their application in a continuous mode. CPC is a chromatographic technique; however, contrarily to the most common chromatographic techniques, CPC does not rely on a solid support with an adsorbent material [151]. Indeed, it is a liquid–liquid chromatography system that requires two immiscible liquid phases acting as stationary and mobile phases, allowing the solutes to partition between the two phases formed [151]. 

The equipment uses a rotator that provides the centrifugal force responsible for holding the stationary liquid phase fixed inside the chromatograph while the mobile phase is pumped through it. CPC grants the maximum recovery of the target compound, allows easy recyclability of the solvents used in the LLE formulation, and presents a reasonable cost due to the total absence of expensive solid chromatographic columns. Thus, CPC minimizes both the environmental footprint and the economic impact of the process. Moreover, a high volume of the stationary phase can be loaded into the equipment reducing sample losses, which could occur by irreversible adsorption to a solid phase, making CPC suitable for industrial applications [151,152]. Organic solvent-dependent systems are the most common choice and have been explored for the purification of carotenoids [153,154]. However, there is an increasing interest in the use of ABSs with alternative solvents in CPC to obtain more efficient and biocompatible systems with a larger range of target molecules [152,155].

Continuous tubular separators (CTS) have been proposed for the continuous operation of aqueous biphasic systems protocols in terms of apparatus geometry and input/output flow rates [156]. Similarly to CPC, in CTS, the geometry of the equipment and ABS composition influence the recovery of the target molecules. The CTS operation is achieved through the injection of phases and the sample into a static mixer coupled to a tubular coalescer and a phase separator as shown in Figure 4. The static mixer is arranged in a spiral shape, composed by bumps in the walls, which cause turbulence of the flow passing through. The prototype proposed by Vázquez-Villegas et al. exploits the tubular reactor approach with a large and adjustable length/diameter ratio to minimize the settling time with reduced manipulation of phases and without the need for external equipment such as centrifuges for phase separation [156,157].

## 6. Conclusions

In this review, we started by focusing on the Portuguese scenario regarding the microalgae business. Then, considering the high commercial potential of pigments and, mainly, the large variety of carotenoids, the downstream processes developed so far for the extraction and purification of these pigments were reviewed. However, today, it is clear that to make microalgae production more competitive, new techniques must be developed, and pigment purification processes must be examined owing to their large influence on the entire cost of the process. For economic viability and expansion of the blue biorefinery sector in Portugal, as well as worldwide, actions must be taken. Some of these actions are summarized below:The microalgae business is only competitive when all the fractions are valorized. For instance, mature infrastructure for algae production for biofuels can be extended to pigment production, further reinforcing a multiproduct bioeconomy which prioritizes an initial recovery of high-value compounds followed by valorization of the remaining fractions (biorefinery concept).Numerous technologies at the laboratory scale could efficiently extract carotenoids from microalgae. Nevertheless, new funding opportunities must be created to allow scalability studies. To this end, supercritical extraction stands out due to the use of nontoxic solvents, although poorly selective, and is being already implemented for industrial application on food production.More work needs to be performed to address the separation and polishing step as the final extract is often of too low purity compared to synthetic pigments.The governmental incentives already implemented (projects in collaboration between universities and industries supported by regional governments and the Recovery and Resilience Plan (RRP) are good examples) requiring fostering of a healthy dialog between the academia and the industry to develop a biorefinery that targets higher-grade compounds needs to continue and be reinforced in the coming years.

The potential for economic improvement, along with sustainable development, sets the framework for further research and establishment of a blue biorefinery.

## Figures and Tables

**Figure 1 molecules-27-02540-f001:**
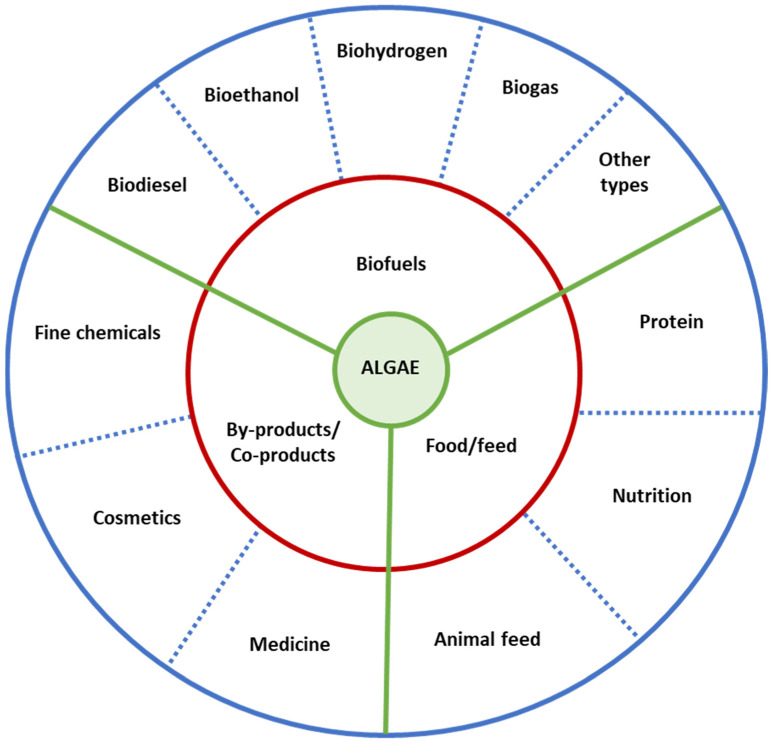
Array of product fields obtained from an algae biorefinery. Adapted with permission from the work of Zhu [21], published by Elsevier, 2015.

**Figure 2 molecules-27-02540-f002:**
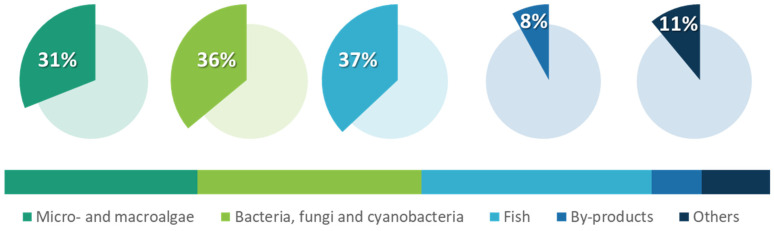
Distribution of the various biomasses used by Portuguese entities. It should be noted that a single entity can exploit multiple biomass sources. Adapted from the Blue Bioeconomy Roadmap for Portugal [30].

**Figure 3 molecules-27-02540-f003:**
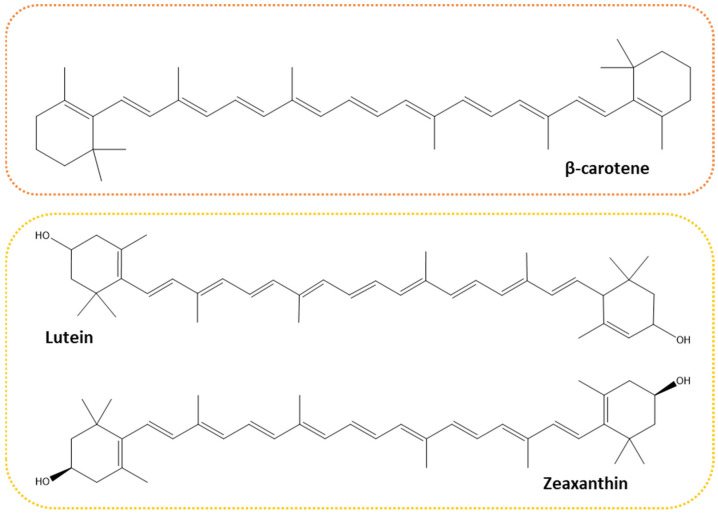
Example of commonly occurring carotenoids in algae. Carotene in the orange outline, xanthophylls—in the yellow outline.

**Figure 4 molecules-27-02540-f004:**
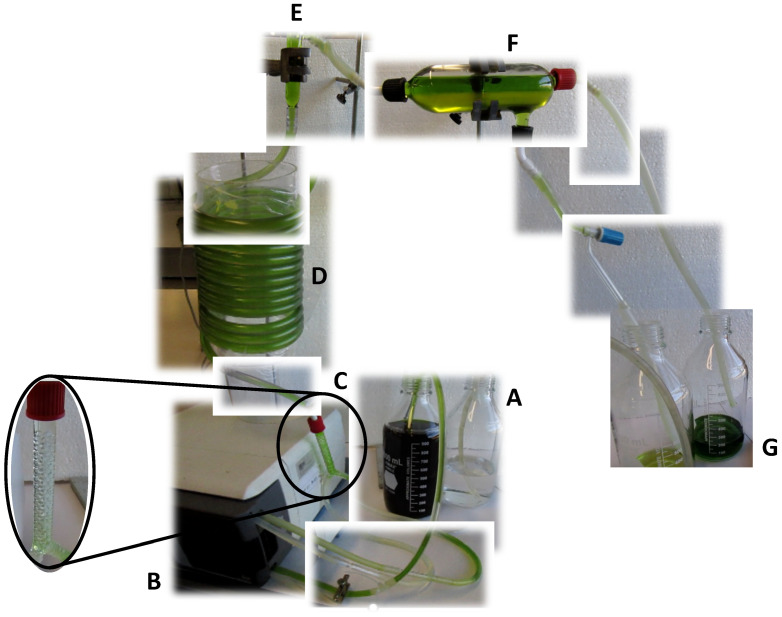
Continuous tubular separator. (**A**) Pre-equilibrated phases and sample solution; (**B**) peristaltic pump; (**C**) phase mixer and turbulence generator; (**D**) continuous tubular separator; (**E**) pre-phase collector to help avoid turbulence in the phase collector; (**F**) phase collector with the interface harvesting port; and (**G**) collector flasks.

**Table 1 molecules-27-02540-t001:** Carotenoids produced by microalgae.

Carotenoid	Microalgae	Other Carotenoids	Concentration	Application Area	Ref.
β-Carotene	*Dunaliella salina*	Zeaxanthin, lutein, α-carotene	10–13% DW	Provitamin A function Colorectal cancer Prevention of acute and chronic coronary syndromes Photoprotection of the skin against UV light Prevention of night blindness Antioxidant property Prevention of liver fibrosis	[11,58,59,60]
	*Chlorella zofingiensis*	Canthaxanthin, astaxanthin	0.9% DW		
	*Spirulina maxima*	Astaxanthin, lutein, β-cryptoxanthin, zeaxanthin, echinenone, oscillaxanthin, myxoxanthophyll	80% TC		
Lutein	*Chlorella pyrenoidosa*	Violaxanthin, loroxanthin, α- and β-carotene	0.2–0.4% DW	Prevention of acute and chronic coronary syndromes and stroke Maintenance of normal visual function Prevention of cataracts Prevention of AMD Prevention of retinitis Prevention of gastric infection by *H. Pylori* Antioxidant properties Anticancer activities	[59,60,61,62,63]
	*Chlorella protothecoides*	–	5.4 mg·g^−1^		
	*Chlorella sorokiniana*	Astaxanthin	5.90 mg·g^−1^		
	*Scenedesmus bijugus*	Astaxanthin	2.9 mg·g^−1^		
Zeaxanthin	*Porphyridium cruentum*	β-carotene	97.4% TC	Prevention of acute and chronic coronary syndromes Maintenance of the normal visual function Prevention of cataracts Prevention of AMD	[59,60,64]

DW—dry weight; TC—total carotenoids.

**Table 2 molecules-27-02540-t002:** Main advantages and disadvantages of several methods of cell disruption.

Cell Disruption Method	Disruption Mechanism	Advantages	Disadvantages	Ref.
Bead mill	Physical deformation by beads against cells	▪ High disruption efficiency ▪ High biomass loading ▪ Good temperature control ▪ Easily scalable	▪ High energy demand ▪ Nonselective procedure ▪ Formation of very fine cell debris	[69,70]
High-speed homogenization	Cavitation and shear	▪ High disruption efficiency ▪ Short contact times	▪ High energy demand ▪ Nonselective procedure	[71]
High-pressure homogenization	Cavitation and shear	▪ Suitable for processing large volumes ▪ Applicable in wet cells ▪ Easily scalable	▪ Nonselective procedure ▪ Formation of very fine cell debris ▪ Not effective to break hard cell walls	[72,73]
Ultrasonication	Cavitation shear force	▪ Easily scalable ▪ Low operational costs ▪ Operated continuously ▪ Can be combined with selective extraction (ultrasound-assisted extraction)	▪ Low cell disruption efficiency for some species ▪ Heat production (cooling necessary)	[65,73]
Microwaves	Temperature increase and molecular energy	▪ Easily scalable ▪ Can be combined with selective extraction (microwave-assisted extraction)	▪ Not ideal for the isolation of volatile compounds ▪ Limited to polar solvents	[74,75]
Pulsed electric field	Short electrical pulses (electroporation)	▪ High disruption efficiency ▪ Low operational costs ▪ Selective ▪ Fast process time	▪ Can promote radical formation and undesired reactions ▪ Depends on the media conductivity ▪ Expensive equipment	[72,73]
Enzymatic lysis	Enzyme substrate interaction	▪ Mild operating conditions ▪ Low energy requirements ▪ Can be combined with other disruption methods ▪ High selectivity ▪ Easily scalable	▪ Expensive ▪ Long process ▪ Product inhibition ▪ Need to know cell composition due to high substrate selectivity ▪ Species-dependent	[65,76]
Osmotic shock	Hypotonic or hypertonic stress	▪ Simple ▪ Low operational costs ▪ Can be used for sensitive biocompounds	▪ Not effective to break hard cell walls ▪ Not suitable for all cell types	[77,78]
Freezing and thawing	Ice crystal formation and perforation	▪ Simple ▪ Easy to implement	▪ Slow ▪ May damage biocompounds ▪ Low yield	[79]
Thermolysis	Heat shock	▪ Independent of the cell type ▪ Easy to implement	▪ Expensive ▪ Possible damage to intracellular components	[75]

**Table 3 molecules-27-02540-t003:** Main advantages and disadvantages of several faction methods and comparison of the methods of extraction of astaxanthin from *Haematococcus pluvialis*.

Solvent	Advantages	Disadvantages	Selected Examples		Ref.
			Conditions	Yield	
Organic	▪ Industrially mature ▪ Easily scalable ▪ Low operational costs ▪ Can be biosourced	▪ Potentially toxic ▪ Nonselective ▪ Volatile ▪ Flammable ▪ Often petroleum-derived	Ethanol:ethyl acetate (1:1 (*v*/*v*), 2 h)	17.34 ± 0.85 (mg·g^−1^)	[82]
			Ethanol:ethyl acetate (1:1 (*v*/*v*), ultrasound-assisted (200 W), 16 min)	27.58 ± 0.40 (mg·g^−1^)	[82]
			Hexane:isopropanol (6:4 (*v*/*v*), ultrasound-assisted, 20 min)	9.7 ± 0.6 (mg·g^−1^)	[83]
Surfactant	▪ Water-based solvent ▪ Low concentrations required	▪ Dificult recovery and recyclability ▪ Potentially toxic	Octanoic acid/ethanol/water ternary mixture (SUPRAS) (ratio of biomass: equilibrium solution: SUPRAS of (1:5:2) (mg:mL:mL))	96 ± 7 (%)	[84]
IL and DES	▪ Selective ▪ Theoretically endlessly recyclable ▪ Nonvolatile ▪ Can be green solvents	▪ Potential be potentially toxic ▪ Viscous ▪ Expensive ▪ Lack of industrial know-how ▪ Complicated polishing	1-ethyl-3methylimidazolium di-butylphosphate (EMIM DBP) (40% (*w*/*w*), at 45 °C, 90 min)	≥70 (%)	[85]
			EMIM-based ILs with HSO_4_^−^, CH_3_SO_3_^−^, (CF_3_SO_2_)_2_*−* anions (6.7% (*v*/*v*), 30 °C, 60 min)	>99.0	[86]
			Thymol:oleic acid (3:1, 6 h, 60 °C, 2.5 wt%)	75 ± 0.7 (%)	[87]
Supercritical	▪ Industrially mature ▪ Lower viscosity ▪ Easy polishing ▪ Green solvents	▪ Higher capital and operational costs ▪ Energy-intensive	SFE-CO_2_ (550 bar, 50 °C, 120 min)	98.6 (%)	[88]

## Data Availability

Not applicable.

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
