# Peer review of "Carotenoid Production from Microalgae: The Portuguese Scenario"

_molecules, 2022, doi:10.3390/molecules27082540_

Round 1

Reviewer 1 Report

Kholany et al. aim to describe and characterise different methods for carotenoids extraction/production. The title of the paper should be better adapted to the text. The descriptions addressed in the different sections are superficial and too general. This is the case, for example, of section 3.2.1 and 3.2.2 where a more detailed and updated bibliography is required. A summary table to show the benefits, yields of the techniques mentioned throughout the manuscript is missed.

Portugal's case is not completely organic with the work, which remains of potential interest, contributing on the contrary to weake its scientific impact.

Author Response

Kholany et al. aim to describe and characterise different methods for carotenoids extraction/production.

The title of the paper should be better adapted to the text.

The authors have better adapted the title of the manuscript.

The descriptions addressed in the different sections are superficial and too general. This is the case, for example, of section 3.2.1 and 3.2.2 where a more detailed and updated bibliography is required.

The authors acknowledge the referee comments and have acted accordingly. Detailed examples were added in each section as well as a Table. Moreover, the bibliography and respective discussion of the review were improved by suggestions obtained from referee #2, as well.

A summary table to show the benefits, yields of the techniques mentioned throughout the manuscript is missed.

The authors acknowledge the referee’s request and added table 3 to show the benefits and yields of the techniques mentioned in the respective chapter.  

Portugal's case is not completely organic with the work, which remains of potential interest, contributing on the contrary to weak its scientific impact.

The authors acknowledge the referee comment and have acted accordingly, also following the suggestions done by referee #2.

Reviewer 2 Report

Manuscript molecules-1622997 is a review work focused on the utilization of microalgae for the production of value-added compounds in the framework of blue biorefineries.

In my opinion, the work deals with a very interesting subject, which fits into the scope of the journal as well as of the special issue. The structure and content correspond to a high scientific standard. The main topics of the article are exhaustively described and discussed. Some points requiring further attention are outlined in the following and in some cases adaptations before acceptance for publication (see attached file).

Author Response

Manuscript molecules-1622997 is a review work focused on the utilization of microalgae for the production of value-added compounds in the framework of blue biorefineries.

In my opinion, the work deals with a very interesting subject, which fits into the scope of the journal as well as of the special issue. The structure and content correspond to a high scientific standard. The main topics of the article are exhaustively described and discussed. Some points requiring further attention are outlined in the following and in some cases adaptations before acceptance for publication.

The authors acknowledge the referee comments and recommendation for acceptance.

  1. In the introduction, the literature survey should be expanded to take into account some parallel routes. I propose here some suggestions (in which portuguese institutions are involved as well). For example, regarding blue biorefinery and microalgae, the following works may be added

- 10.1016/j.biortech.2019.122509

- 10.3390/molecules25153406

An important work related to carotenoids from microalgae is the following review

- 10.3390/md9040625

Finally, regarding the energetic valorization, some works may be added as well, especially regarding hydrothermal liquefaction, which is one of the most studied in the last years

- 10.1021/acsami.7b16185

- 10.1016/j.bcab.2021.102045

Answer 1 - The authors acknowledge the referee suggestions of bibliography. The revised version of this manuscript was improved not only by adding the references indicated by the referee but also by using them to improve the discussion.

  1. In the introduction, the authors outline the importance of maximizing the recovery of compounds from microalgae. However, there is a lack of information in the main text regarding these points. Some evaluations on that (e.g., citing techno-economical studies) are strongly suggested.

Answer 2 - The authors acknowledge the referee comments and acted accordingly by adding a section in the introduction dedicated to exposing some techno-economic analysis for carotenoids.

  1. The authors pointed out that the strain Dunaliella salina lacks a cell wall, so I assume that the “cell disruption” step could be avoided. If this is true, how does it affect the economics of the plant?

Answer 3- The authors acknowledge the referee comment. Indeed, because Dunaliella salina lacks a rigid cell wall, it is more fragile and easily ruptured, avoiding the energy cost inputs required to enable efficient cell breaking and make carotenoids readily available for solvent-mediated extraction. Thus, the pigment recovery might be achieved at lower costs when compared to certain other microorganisms.

  1. Proof-reading should be carried out to fix typos and grammatical mistakes.

Answer 4- The authors acknowledge the referee comments and have acted accordingly.

  1. I would suggest the revision of the introduction section, that is sometimes limping. You start with biorefinery and microalgae, then you move to the “microalgae biorefinery” explaining the steps, then it seems to me that you go back introducing again the microalgae “Microalgae are recognized worldwide as a plentiful/valuable source of carbon…”.

Answer 5 - We thank the reviewer for pointing this potential source of confusion. In this context, the manuscript was revised and chapter 2, “The concept of (blue) biorefinery” section updated.

  1. The sentence in row 81-84 should be rephrased.

Answer 6 - We thank the reviewer for pointing this potential source of confusion. This was clarified in the manuscript with the following correction to the sentence:

“In this work, we will focus on the different characteristics of the carotenoids produced by microalgae and their applications and will assess what is currently being done in terms of bioprocesses.”

  1. To avoid misleadings, the title of chapter 3 should be modified appropriately to include also the cell disruption step.

Answer 7- The authors acknowledge the referee comments and have acted accordingly.

Reviewer 3 Report

The manuscript ID molecules-1622997 entitled “Microalgae Valorization in Portugal under the Concept of Blue Biorefinery” is an interesting study. The Authors analysed the available literature related to the use of microalgae biomass in Blu biorefinery concept. The use of 130 literature items is impressive bot in my opinion, the topic has not been sufficiently described and should be completed.

The Authors properly present the current scientific achievements in the field of microalgae role in biorefinery. The authors effectively try to analyse and summarize the current state of advancement of this type of technology and the possibilities of their application for carotenoids obtaining. The manuscript is written in the correct language. The presented content corresponds to the Molecules journal profile.

My critical remarks below:

  1. In my opinion, the title is too general and therefore misleads the reader. Valorization of microalgae biomass concerns, in principle, only carotenoids and methods of their acquisition. The Blue Biorefinery concept is only presented in a perspective and future-oriented way. The title should fit the content of the manuscript. Please correct.
  2. Abstract should be corrected. It lacks conclusions, observations, and critical opinions of the authors about technologies based on microalgae. Review work should lead to reflection, summing up, identifying potential and possible threats. Based on the collected material, the authors should develop their opinion and present it to the reader. Review work cannot be just a thoughtless compilation of knowledge available to everyone. It have to be improve.
  3. The most important thing is missing from the keywords. I mean: carotenoids. It need to be supplemented.
  4. In my opinion layout and the extracted chapters are not logical. In the manuscript, the first chapter should be Introduction. The authors omitted or presented only in slogan many important issues. Please explain the concept of Blue Biorefinery more broadly or does it apply only to biomass of microalgae obtained from natural waters, excluding controlled cultivation? However, the content presented in the first chapter shows that it is not.
  5. Why exactly did the authors focus on carotenoids? Is this a specific direction for the production and use of microalgae biomass in Portugal? Please explain.
  6. Please explain whether chapter "3 Extraction and purification techniques" covers carotenoids or all other valuable components derived from microalgae biomass.
  7. I think chapter “4 Microalgae biorefinery in Portugal: Past, present, and future” should be chapter 2 after chapter “1. Introduction ".
  8. First of all, technologies for the production of microalgae biomass become efficient and profitable when the breeding medium is relatively cheap, preferably prepared on the basis of sewage or other liquid waste: https://doi.org/10.3390/w12072071, https://doi.org/10.3390/pr8050517, https://doi.org/10.3390/w12010106. This approach fits the essence directly into the issues of the circular economy and the bio-refinery concept. It has to be mention and supplemented in Introduction section and also in final summary
  9. The important area of microalgae biomass use has not been sufficiently mentioned in the aspect of the Blue Biorefinery concept regarding CO2 capture and utilisation. It is one of the basic and future directions of the application of technologies based on microalgae biomass: https://doi.org/10.3390/atmos12081031, https://doi.org/10.3390/en13020413, https://doi.org/10.3390/app112411931. This topic should be considerably extended.
  10. Information on the possible energetic use of microalgae biomass, including both for the production of various types of biofuels (biogas, biohydrogen, bioethanol, biodiesel and other), also requires extension and a comprehensive approach: https://doi.org/10.3390/app9224793, https://doi.org/10.3390/en14196025, https://doi.org/10.3390/su13168797, as well as emissions from their use: https://doi.org/10.3390/ijerph17113896, https://doi.org/10.3390/en12132546, https://doi.org/10.3390/atmos12091099 .
  11. The authors should also pay attention and analyse the strengths and weaknesses of microalgae technologies from a Portuguese perspective: https://doi.org/10.3390/app9224793, https://doi.org/10.3390/md19060319, https://doi.org/10.3390/su12239980, https://doi.org/10.3390/en14051416,
  12. The manuscript lacks the final summary and conclusions, which should include the critical views of the authors.
  13. The manuscript is not written according to the Molecules journal template. Information is missing: Author Contributions, Conflicts of Interest, etc.

Author Response

The manuscript ID molecules-1622997 entitled “Microalgae Valorization in Portugal under the Concept of Blue Biorefinery” is an interesting study. The Authors analysed the available literature related to the use of microalgae biomass in Blu biorefinery concept. The use of 130 literature items is impressive but, in my opinion, the topic has not been sufficiently described and should be completed.

The Authors properly present the current scientific achievements in the field of microalgae role in biorefinery. The authors effectively try to analyse and summarize the current state of advancement of this type of technology and the possibilities of their application for carotenoids obtaining. The manuscript is written in the correct language. The presented content corresponds to the Molecules journal profile.

The authors acknowledge the referee comments and recommendation for acceptance.

My critical remarks below:

  1. In my opinion, the title is too general and therefore misleads the reader. Valorization of microalgae biomass concerns, in principle, only carotenoids and methods of their acquisition. The Blue Biorefinery concept is only presented in a perspective and future-oriented way. The title should fit the content of the manuscript. Please correct.

Answer 1- The authors have better adapted the title of the manuscript.

  1. Abstract should be corrected. It lacks conclusions, observations, and critical opinions of the authors about technologies based on microalgae. Review work should lead to reflection, summing up, identifying potential and possible threats. Based on the collected material, the authors should develop their opinion and present it to the reader. Review work cannot be just a thoughtless compilation of knowledge available to everyone. It have to be improve.

Answer 2- The authors thank and acknowledge the referee comments. As suggested by the reviewer, we restructured the abstract where we dive into our critical views over the work.

  1. The most important thing is missing from the keywords. I mean: carotenoids. It need to be supplemented.

Answer 3- The authors acknowledge the referee comments and have acted accordingly.

  1. In my opinion layout and the extracted chapters are not logical. In the manuscript, the first chapter should be Introduction. The authors omitted or presented only in slogan many important issues. Please explain the concept of Blue Biorefinery more broadly or does it apply only to biomass of microalgae obtained from natural waters, excluding controlled cultivation? However, the content presented in the first chapter shows that it is not.

Answer 4 - The authors acknowledge the referee comments and have made changes demanded by the referee.

  1. Why exactly did the authors focus on carotenoids? Is this a specific direction for the production and use of microalgae biomass in Portugal? Please explain.

Answer 5- The authors acknowledge the referee comment. Within our research and line of work we focus on pigments as value added compounds, targeting their recovery. The reason behind our choice regarding the focus on carotenoids is not only related with the fact that we recognize the value of these colorants, but also considering the demands of the companies we are working with in the topic. Nevertheless, we are aware of the importance of other pigments and metabolites present in microalgae and their high commercial potential and, in this sense, we have a second review paper under preparation focusing other class of pigments, the phycobiliproteins.

Finally, and considering the potential commercial impact of carotenoids, the authors are also critical about the importance of developing platforms for the recovery of carotenoids in Portugal – which has been expanded in the new subtopic - conclusions.

  1. Please explain whether chapter "3 Extraction and purification techniques" covers carotenoids or all other valuable components derived from microalgae biomass.

Answer 6 – The authors acknowledge the referee question. We need to start by clarifying that the major scope of this review are the carotenoids, however, it is impossible to talk about the biorefinery concept associated to microalgae without mention (briefly) the other components composing the microalgae cells.

  1. I think chapter “4 Microalgae biorefinery in Portugal: Past, present, and future” should be chapter 2 after chapter “1. Introduction ".

Answer 7- The authors acknowledge the referee comments and have acted accordingly.

  1. First of all, technologies for the production of microalgae biomass become efficient and profitable when the breeding medium is relatively cheap, preferably prepared on the basis of sewage or other liquid waste: https://doi.org/10.3390/w12072071https://doi.org/10.3390/pr8050517https://doi.org/10.3390/w12010106. This approach fits the essence directly into the issues of the circular economy and the bio-refinery concept. It has to be mention and supplemented in Introduction section and also in final summary

Answer 8- The authors acknowledge the referee comments and have acted accordingly.

  1. The important area of microalgae biomass use has not been sufficiently mentioned in the aspect of the Blue Biorefinery concept regarding CO2 capture and utilisation. It is one of the basic and future directions of the application of technologies based on microalgae biomass: https://doi.org/10.3390/atmos12081031https://doi.org/10.3390/en13020413https://doi.org/10.3390/app112411931.This topic should be considerably extended.

Answer 9- The authors acknowledge the referee comments. As suggested by the reviewer, we are aware of the importance of CO2 sequestration by using microalgae, as a way of fighting climate changes by decreasing the greenhouse gas emissions. However, the scope of the present review is mainly focusing the downstream processing of the biomass, and extensive review on the upstream step is, in our opinion, out of scope. Moreover, we also want to call the attention of the reviewer that, we have avoided not only to expand the upstream part of the processes, including the cultivation of microalgae using wastes or by reutilizing CO2 from other industries, because we are aware of the legislation problem/gap associated. For the use of biomass or biobased compounds derived from biomass produced by reutilizing CO2 from other industries, it is necessary to guarantee that the gases emitted by the industries fulfils the level of heavy metals (just one of various examples) approved by legislation. Thus, to avoid this discussion, which in our opinion, is out of the scope of this review, we have opted by “not sufficiently mentioning” these topics, to use the referee own words.  

  1. Information on the possible energetic use of microalgae biomass, including both for the production of various types of biofuels (biogas, biohydrogen, bioethanol, biodiesel and other), also requires extension and a comprehensive approach: https://doi.org/10.3390/app9224793https://doi.org/10.3390/en14196025https://doi.org/10.3390/su13168797, as well as emissions from their use:https://doi.org/10.3390/ijerph17113896https://doi.org/10.3390/en12132546https://doi.org/10.3390/atmos12091099

Answer 10- The authors acknowledge the referee comments. As suggested by the reviewer, we made readers aware of the relevance and extent to which bio-energy solutions are applied. This was achieved through referral to several reviews on the topic, this allowing the document to naturally progress to the downstream processing of pigments.

  1. The authors should also pay attention and analyse the strengths and weaknesses of microalgae technologies from a Portuguese perspective: https://doi.org/10.3390/app9224793https://doi.org/10.3390/md19060319https://doi.org/10.3390/su12239980https://doi.org/10.3390/en14051416

Answer 11- The authors acknowledge the referee comments. As suggested by the reviewer, further information on the stakeholders’ perspective was described and two of the references deemed most relevant were added in the revised version of the manuscript.

  1. The manuscript lacks the final summary and conclusions, which should include the critical views of the authors.

Answer 12- The authors thank and acknowledge the referee comments. As suggested by the reviewer, we added a conclusion chapter where we dive into our critical views over the work.

  1. The manuscript is not written according to the Molecules journal template. Information is missing: Author Contributions, Conflicts of Interest, etc.

Answer 13- The authors acknowledge the referee comments and have acted accordingly.

Round 2

Reviewer 1 Report

The authors' work to improve the review is appreciable, but the question remains about the scientific impact of this work and its suitability for publication in a journal with a significant impact such as Molecules. 

Reviews have already been published in the literature that describe the topic discussed in the present paper in more detail and depth:

Liu et al., 2021. Carotenoids from fungi and microalgae: A review on their recent production, extraction, and developments. https://doi.org/10.1016/j.biortech.2021.125398

Ren et al., 2021. Carotenoid Production from Microalgae: Biosynthesis, Salinity Responses and Novel Biotechnologies. https://doi.org/
10.3390/md19120713

Reviewer 3 Report

Thanks to the authors for the significant improvement of the manuscript. In my opinion, the work can be published in its current form.